# A Comparative Investigation on Structural and Chemical Differences between the Pith and Rind of Sunflower Stalk and Their Influences on Nanofibrillation Efficiency

**DOI:** 10.3390/polym14050930

**Published:** 2022-02-25

**Authors:** Lingyan Zhang, Wenting Ren, Fangqingxin Liu, Linmin Xia, Xiaomei Wu, Rilong Yang, Yan Yu, Xuexia Zhang

**Affiliations:** 1College of Material Engineering, Fujian Agriculture and Forestry University, Fuzhou 350108, China; zly940255600@163.com (L.Z.); 18856022706@163.com (W.R.); kuluomigongzhu@163.com (F.L.); xialinmin2016@163.com (L.X.); xiaomeizi163@126.com (X.W.); rilongyang@foxmail.com (R.Y.); 2Agricultural-Engineering Institute, Fujian Academy of Agricultural Sciences, Fuzhou 350003, China

**Keywords:** sunflower stalk, chemical composition, cell structure, cellulose nanofiber, nanofibrillation

## Abstract

The structure and chemical composition of cell walls play a vital role in the bioconversion and utilization of plants. In the present study, the cell wall structure and chemical composition of pith and rind from sunflower stalks were compared and correlated to their nanofibrillation efficiency with ultrasonic treatment. Mild chemical pretreatment using 1% or 4% NaOH without any bleaching process were applied prior to ultrasonication nanofibrillation. Significant structural and chemical differences were demonstrated between the pith and rind, with the former exhibiting a much lower lignin and hemicellulose contents, higher pectin, much looser cell structure and higher cell wall porosity than the latter. Alkaline treatment alone was sufficient to eliminate most of the hemicellulose and pectin from stalk pith, whereas only partial removal of hemicellulose and lignin was achieved for the woody rind part. After 30 min of ultrasonic treatment, the stalk pith exhibited fully defibrillated fibrils with a continuous and entangled micro/nanofibrillated network, whereas numerous micron-sized fiber and fragments remained for the rind. The results indicated that stalk pith is less recalcitrant and easier to be fibrillated with ultrasonication than rind, which must be correlated to their distinct differences in both structure and chemical composition.

## 1. Introduction

As the most abundant and renewable biopolymer on Earth, cellulose has drawn considerable attention in the material field for the production of environmental-friendly materials [1]. Cellulose nanofiber (CNF) has attracted extensive interest owing to its remarkable properties such as high tensile modulus and stiffness, large surface area, low density and biocompatibility [2] and thus have been used in advanced materials especially for biomedical research, electronic devices and food packaging, etc. [3,4]. However, the presence of lignin in lignocellulosic biomass is the main cause of biomass recalcitrance during separation processes. Lignin is a heterogeneous, cross-linked three-dimensional phenyl-propane polymer, acting as a protective barrier for reducing plant cell permeability and increasing resistance against microbial attack [5,6]. Moreover, lignin content is an important factor in the processing of CNF, so to produce CNF a precursor with low lignin content is preferable [7]. Tanpichai et al. [8] reported that production of CNF from water hyacinth with low lignin content (4.1%) could avoid tedious and energy-consuming treatment procedures. Thus, a raw material with a limited lignin content would represent a promising alternative source for the production of CNF to wood or other plants with higher lignin content.

Sunflower (*Helianthus annuus* L.), an annual herbaceous plant, is the third largest source of vegetable oil worldwide after soybean and palm, and more than 27,000 ha of sunflower were harvested globally in 2019 [9,10]. Consequently, the amount of sunflower stalks shredded during flower harvesting is increasing year by year. The stalk consists of rind (peripheral brown part of stalk) and pith (central white part of stalk), which account for approximately 90% and 10% of the stalk by weight, respectively [11], as shown in Figure 1. Historically, stalk rind has been investigated primarily to produce paper [12], thermoplastic composites [13], bioethanol [14] and energy fuel [15]. In general, the pith is considered a waste and removed before stalk utilization [12]. Very recently, stalk pith has been receiving increasing attentions in solar evaporators [16,17], and feedstock for the coproduction of pectin and glucose [18]. Currently, unlike stalk rind, whose anatomy and cell wall ultrastructure has been investigated thoroughly [11], information regarding stalk pith and its effects on bioconversion and utilization has not been reported.

Taking this into consideration, this study systematically characterized the chemical composition and microstructure characteristics of pith and rind and evaluated their performances in the chemical pretreatment and nanofibrillation process of CNF production. This study aimed to elucidate the cell wall structural and compositional factors that contribute nanofibrillation behavior of rind and pith, and to provide insights for better utilization of sunflower stalks.

## 2. Materials and Methods

### 2.1. Materials

Sunflower stalks were supplied by the Yuanfeng Biomass Thermal Power Plant, located in Ordos, Inner Mongolia, China. The sunflower stalks were separated into the pith and rind, respectively, and both were ground into powder with a grinding machine and then filtered through stainless steel mesh (10–30 mesh). Sodium hydroxide (NaOH, ≥98%) was obtained from Aladdin Chemistry Co., Ltd. (Shanghai, China) and used as received. Deionized water was used in all experiments.

### 2.2. Preparation of Cellulose Nanofiber

A facile strategy using mild alkaline treatment followed by ultrasonic nanofibrillation was applied to prepare cellulose nanofiber, which is illustrated in Figure 2. Firstly, the pith and rind were soaked in NaOH (1 wt.% or 4 wt.%) solution at 90 °C for 1 h, and labeled as P-1, P-4, R-1 and R-4, respectively. Subsequently, the samples were soaked in distilled water (0.5 wt.%) and placed in an ultrasonic generator (JY99-IIDN, Ningbo Scientz Biotechnology Co., Ltd., Ningbo, China) equipped with a 25-mm titanium probe. The ultrasonic processing was conducted at output power of 540 W for 30 min, resulting in the CNF suspensions that named as P-1-U, P-4-U, R-1-U and R-4-U, respectively.

### 2.3. Characterizations

The sunflower stalk was cut into blocks, and the height and area of the samples were measured by a Vernier caliper and the image J software, respectively. After weighing the whole stalks, the pith was carefully separated from them manually and weighed. The mass and volume of all sample were then analyzed, after which their density in the sunflower stalks was calculated. The analyses were performed in sextuplicate.

Morphology of samples was investigated using a FEI Nova NanoSEM 230 instrument (FEI, Hillsboro, OR, USA) at an accelerating voltage of 5 kV and a working distance of 5 mm. Prior to SEM observation, the samples were sputter-coated with platinum using a vacuum sputter coater. The morphology of supernatant suspension of samples was imaged using a transmission electron microscope (TEM, JEM-1400plus, JEOL, Tokyo, Japan). Diameter measurements were performed on 100 randomly selected CNFs using the Image J software. Brunauer-Emmett-Teller (BET) specific surface area and pore volume were measured using an ASAP 2460 apparatus (Micromeritics, Norcross, GA, USA) through nitrogen adsorption at −196.15 °C. Samples were degassed at 80 °C for 10 h under vacuum. The FTIR spectra of samples were characterized by a VERTEX 70 FTIR spectrometer (Bruker, Karlsruhe, Germany). The spectrum was measured in ATR mode in the range 400–4000 cm^−1^ with a resolution of 4 cm^−1^. Thermal stability of samples was evaluated using a STA449F3 thermal analyzer (Netzsch, Selb, Germany) at a heating rate of 10 °C/min in nitrogen atmosphere.

The carbohydrates and lignin, extractive, ash contents of pith and rind were determined according to the method proposed by National Renewable Energy Laboratory (NREL, Golden, CO, USA). Monomer sugars including glucose, xylose, arabinose and galactose were measured using ion chromatography (IC, DIONEX ICS-5000+, Thermo Scientific, Waltham, MA, USA). Pectin contents were measured according to a Chinese national standard (GB 10742-08). The measurements were performed in triplicate, and the averaged valued were calculated.

## 3. Results and Discussion

### 3.1. Morphology and Chemical Composition of Pith and Rind

As shown in Table 1, the stalk rind is much denser, with an air-dried density of 318 mg/cm^3^, while the pith is lightweight, with an extremely low density of 27 mg/cm^3^. Such significant differences in density are related to their distinct cell wall type and micro-structure, as shown in Figure 3. The SEM images showed that the pith contains entirely parenchyma cells (Figure 3a,c), while the rind contains mainly fibers, which are accompanied by a certain number of vessels and parenchyma cells (Figure 3d,f). Parenchyma cells have thin cell walls with a large lumen, whereas the fibers had thick cell walls with a small lumen. Parenchyma cells in pith exhibited an intriguing honeycomb architecture with approximately tetrahedral-like shape. The diameter of lumina was around 119 μm and cell wall thickness was around 0.87 μm. Such unique porous structure with huge lumina has been reported to endow the stalk pith with highly efficient water transportation when used as solar evaporators [17]. Presence of cavities in cell corners were observed in pith (arrow in Figure 3b), and similar phenomenon were also observed in bamboo [19]. Rind is composed of numerous thick-walled fibers with the average lumina diameter of about 10 μm and cell wall thickness of about 2.93 μm.

Apart from morphological difference, pith and rind also differ dramatically in cell wall porosity. The N_2_ sorption isotherms and pore size distributions of pith and rind are shown in Figure 4. Both the isotherm of pith and rind were intermediate, between type II and type IV according to IUPAC classification, which means that the presence of slit-shaped mesopores in both samples [20]. Particularly, pith shows apparent hysteresis loops at medium relative pressure, suggesting the presence of abundant micropores [21]. As shown in Figure 4b, the pore size distribution in pith and rind exhibited a wide range of micro- and mesopores from 1 to 13 nm. The pith exhibited one narrow and sharp peak around 1.3 nm within the scope of micropores and intense peak around 11 nm in the scope of mesopores, which was quite different from the rind with small peak around 2 nm. The results indicated that there were more micro- and mesopores in pith than that in rind, which was confirmed by the much higher value of specific surface area (1.77 vs. 0.73 m^2^/g) and pore volume (0.006 vs. 0.003 cm^3^/g) reported in pith as compared to rind. Porosity of plant cell wall plays a vital role during biomass pretreatments and higher cell wall porosity normally results in a more accessible cell wall for chemical agents [22,23]. Thus, the significant porosity differences between pith and rind should influence their behavior during chemical pretreatment.

The chemical composition of pith and rind was shown in Figure 5. The pith had a comparable glucose content with rind (28.81% vs. 32.54%), indicating that both pith and rind are potential cellulose resources for valued-added products. However, the pith had significantly lower xylose and lignin content, 1.69%, and 1.53% respectively, whereas corresponding values for rind were 15.49% and 17.18%. Besides, the principal hemicellulose of the rind is xylose (15.49%), followed by galactose (0.78%) and arabinose (0.03%), whereas the hemicellulose monosaccharide content of pith is very low. It is noteworthy that the pith has extremely lower lignin which distinguishes itself from the other stalk pith, e.g., pith from corn stalk (12.6% lignin, [24]), pith from sugarcane bagasse stalk (20.3% lignin, [25]), and pith from *Miscanthus×giganteus* stalk (19.4% lignin, [26]). 

Previous studies suggested that lignocellulosic materials with higher amount of lignin shows more recalcitrance and resistance against cellulose degradation than lignin-poor feedstocks [5,27]. It should be mentioned that pith is rich in pectin compared to rind (23.23% vs. 5.63%). This may be because parenchyma cell walls from pith is composed of primary cells, and pectin is an important component of the primary cell wall which can influence the porosity of the cell walls and morphogenesis of plant [18].

### 3.2. Alkaline Pretreatment of Pith and Rind

Figure 6 depicts FTIR spectra of the raw and pretreated samples. As for pith, the peaks at 1733 and 1250 cm^−1^, 1512 cm^−1^ and 962 cm^−1^, assigned to hemicellulose, lignin and pectin, respectively, totally disappeared [18,28,29], indicating that they were completely eliminated after 1% or 4% alkali treatment. In contrast, the peak at 1511 cm^−1^ related to lignin was also found in the pretreated R-1 and R-4 samples, indicating that the alkaline pretreatment had a limited delignification effect on the rind. Although the band at 1738 and 1250 cm^−1^ disappeared after mild alkali treatment, a new peak at 1457 cm^−1^ related to xylan rings was found in the R-1 and R-4 samples, which might be a consequence of a partial removal of hemicellulose. These results indicated that mild alkali was insufficient to remove lignin and hemicellulose from the rind. Similar phenomena in cellulosic materials such as water hyacinth and sugarcane bagasse after chemical treatments have also been reported by other researchers [8,28]. Previous studies showed a combined homogenization-high intensity ultrasonication process for effective individualization of cellulose micro-nano fibers from rice straw for removal of hemicellulose and lignin was desirable prior to mechanical nanofibrillation into CNF, and the effective removal of lignin is rather tough, which is usually achieved via a two-step chemical treatment combining alkaline treatment and bleaching process [30]. Therefore, considering the high lignin content in the raw rind materials, mild alkaline treatment alone can only achieve partial removal of the hemicellulose and lignin.

Figure 7 shows the morphology changes of pith and rind after alkaline pretreatment. Cell wall fragments of parenchyma cells were clearly observed in the pretreated P-1 and P-4 samples, indicating the pith was separated into individual parenchyma cells (Figure 7a,b). This was due to the fact that pectin, which functions as a cell wall adhesive, was completely removed during alkaline treatment [31,32]. However, Figure 7c,d show that cell walls from rind were still bound together after alkaline treatment, i.e., the fibers were not completely individualized after 1% or 4% alkaline treatment. The results were consistent with the above mentioned FTIR results, conforming that the mild alkaline treatment alone applied in this study was not sufficient to completely remove hemicellulose and lignin which act as binder for adjacent fibers from rind samples [33].

Figure 8 shows the TGA and DTG curves of raw and alkaline pretreated samples. In all cases, a small weight loss was found between 30 and 100 °C due to the evaporation of water from the materials [29]. The starting decomposition temperature occurred at 221 °C for raw pith, and shifted to 255 °C for P-1, further shifting to 261 °C for P-4, which implied that the alkaline treatment effectively removed hemicelluloses, lignin and pectin within pith (Table 2). A similar phenomenon was observed for the raw and pretreated rind samples, but a higher initial decomposition temperature was found in all the rind samples compared to corresponding pith samples due to the presence of lignin. The thermal stability of pretreated samples was appreciably improved compared to that of the original samples due to the partial removal of non-cellulosic materials. The pretreated pith and rind have similar thermal behavior, and the T_30_ and T_max_ of pretreated rind are only slightly higher than those of pretreated pith. The results shows that alkaline treatment had positive effects on the thermal stability of both stalk pith and rind.

### 3.3. Nanofibrillation Behavior of Pith and Rind

After pretreatment, the samples were subjected to 30-min ultrasonic treatment to prepare CNF thorough ultrasonic cavitation. In order to evaluate the degree of nanofibrillation of samples, the dispersion of the CNF in water was compared and shown in Figure 9. Raw pith floated on water, whereas raw rind sank at the bottom of the glass bottle, which was consistent with the above mentioned density results. Both CNF suspensions obtained from pith were stable and no precipitates were observed (Figure 9a), which indicated that the size of the CNF particles was small enough and they became entangled which is responsible the well dispersed state. However, clear sedimentation was observed in R-U-1 and R-U-4 suspensions after ultrasonic treatment (Figure 9b), suggesting an inadequate nanofibrillation time for ultrasonic treatment in rind samples, i.e., the rind sample was not easily nanofibrillated into CNF.

The dispersive state of pith and rind suspension is mainly determined by their morphologies, as shown in Figure 10. 

The pith presented fully defibrillated fibrils with a continuous and entangled micro/nanofibrillated network shown in P-1-U (Figure 10a) and P-4-U (Figure 10b). The diameters of fibrillated fibril in P-4-U were smaller than those of P-1-U. In contrast, stalk rind appeared to be more recalcitrant towards ultrasonication fibrillation as numerous micron-sized fiber and fragments with diameters of 2~4 μm are prevalent both in the R-1-U and R-4-U samples (Figure 10c,d). In addition, the occurrence of spherical lignin particles could be observed, likely due to lignin aggregation resulting from its aromatic structure and amphiphilic nature [34]. Therefore, only alkali treatment and ultrasound are insufficient to fibrillate rind. This may be attributed to the presence of non-cellulosic polysaccharides and lignin, which may cause difficulties during nanofibrillation [8,35].

TEM was used to investigate the diameter distribution of the CNF suspension supernatants obtained from pith and rind (Figure 11). One can note that the morphology of CNF from pith and rind obtained by ultrasonic treatment is quite similar, regardless of the structure and chemical composition of the raw materials. The average diameter of CNF was about 7 nm both in P-1-U and P-4-U samples, which is slightly smaller than those of R-1-U and R-4-U. The CNF diameters in this study are comparable to those of particles extracted from other plants such as wood, sugarcane bagasse, and wheat in terms of the width [28,36,37].

Based on the above discussion, stalk pith is more favorable for fibrillation by ultrasonication than rind, which mainly benefits from the following aspects. Firstly, pith contains entirely parenchyma cells with thin walls and large lumen, while rind contains mainly fibers with thick cell walls and small lumen. Therefore, on the one hand, the stiff fibers will exhibit much stronger mechanical stability to ultrasonication as compared the pith. On the other hand, it was more difficult for the alkaline solution to penetrate into fibers compared with parenchyma cells. Secondly, the lignin-poor pith is less recalcitrant and resistant against cellulose degradation than rind. Moreover, the cell wall of stalk pith possesses higher porosity than rind, making the cell walls more accessible to chemical agents and enhancing the efficiency of ultrasonic cavitation. All the above features suggest that stalk pith is easier to fibrillate and more suitable than rind for CNF production in terms of efficiency and energy consumption.

## 4. Conclusions

In this study, the cell wall structures and chemical composition of pith and rind from sunflower stalks were compared and correlated to their nanofibrillation efficiency. The results revealed significant structural and chemical differences between pith and rind, with much lower lignin and hemicellulose contents, higher pectin, much looser cell structure and higher cell wall porosity for the former. FTIR spectroscopy indicated that mild alkaline treatment could remove hemicellulose and pectin well from pith, while is was insufficient to remove all the lignin and hemicellulose from rind. A delignification process would be required to improve the nanofibrillation efficiency of the stalk rind. The results indicted that after 30 min of ultrasonic treatment, pith suspensions can achieve homogenous dispersion, but clear sedimentation existed in the suspensions of rind. The results revealed that the stalk pith is much less recalcitrant and easier to fibrillate compared to rind. It is envisioned that the waste stalk pith, with its advantages of high cellulose content, low lignin content, and a porous structure may be a new candidate to prepare functional cellulose-based materials and bioinspired structures.

## Figures and Tables

**Figure 1 polymers-14-00930-f001:**
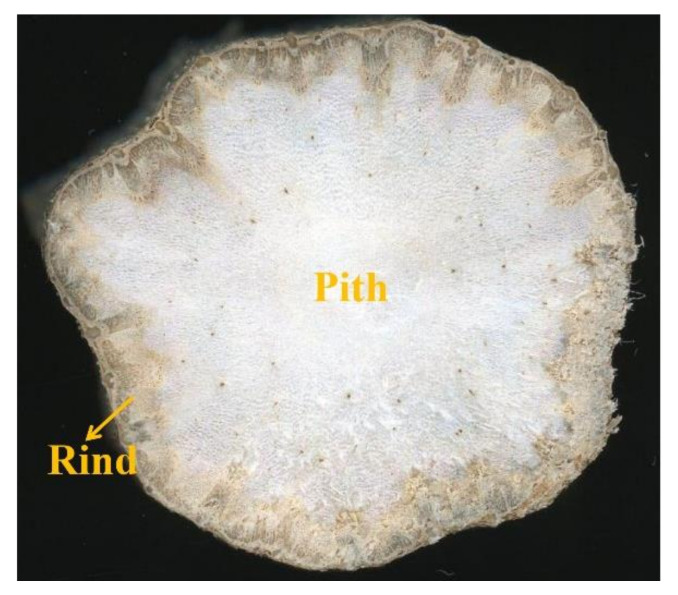
Cross-section of sunflower stalk.

**Figure 2 polymers-14-00930-f002:**
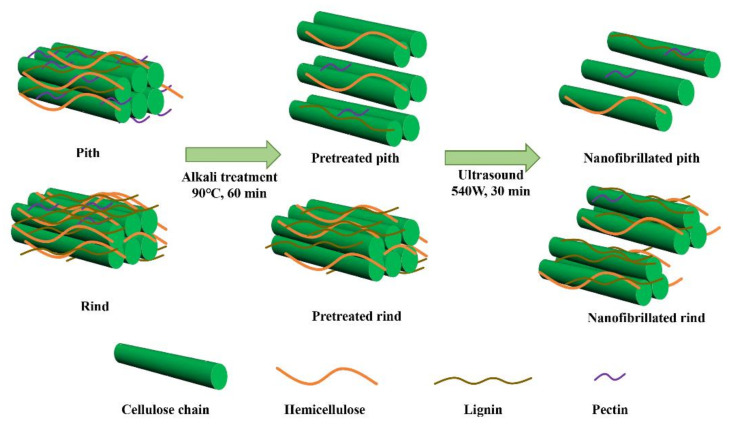
A schematic flow diagram of sunflower pith and rind for producing CNF.

**Figure 3 polymers-14-00930-f003:**
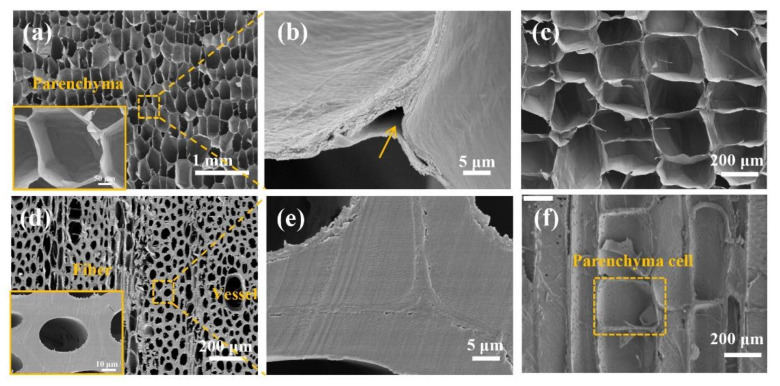
Microscopic structure of sunflower stalk. the transverse cross-section of pith (**a**,**b**) and rind (**d**,**e**); the longitudinal cross-section of pith (**c**) and rind (**f**).

**Figure 4 polymers-14-00930-f004:**
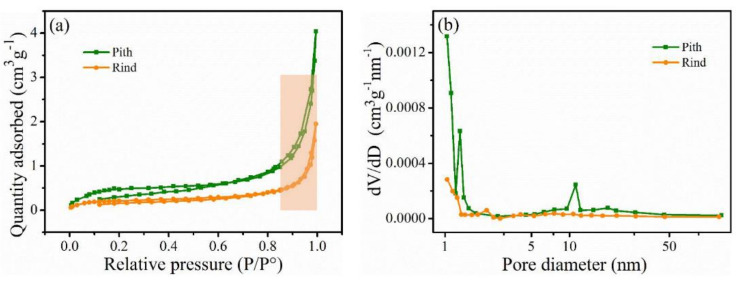
N_2_ adsorption-desorption isotherms (**a**) and pore size distributions (**b**) of the pith and rind.

**Figure 5 polymers-14-00930-f005:**
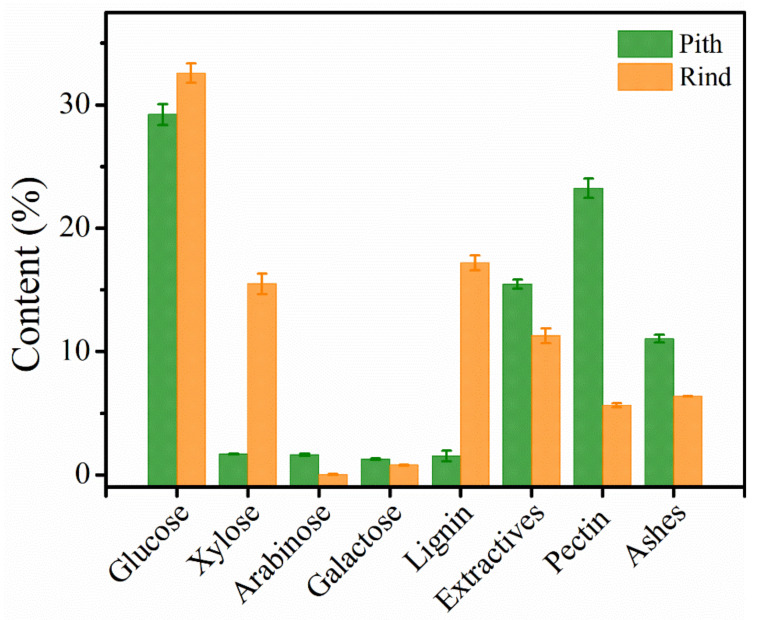
Chemical composition of pith and rind.

**Figure 6 polymers-14-00930-f006:**
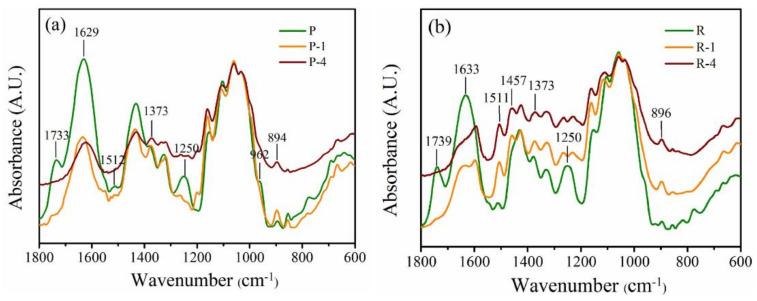
FTIR spectra of rind (**a**) and rind (**b**) with alkali treatments.

**Figure 7 polymers-14-00930-f007:**
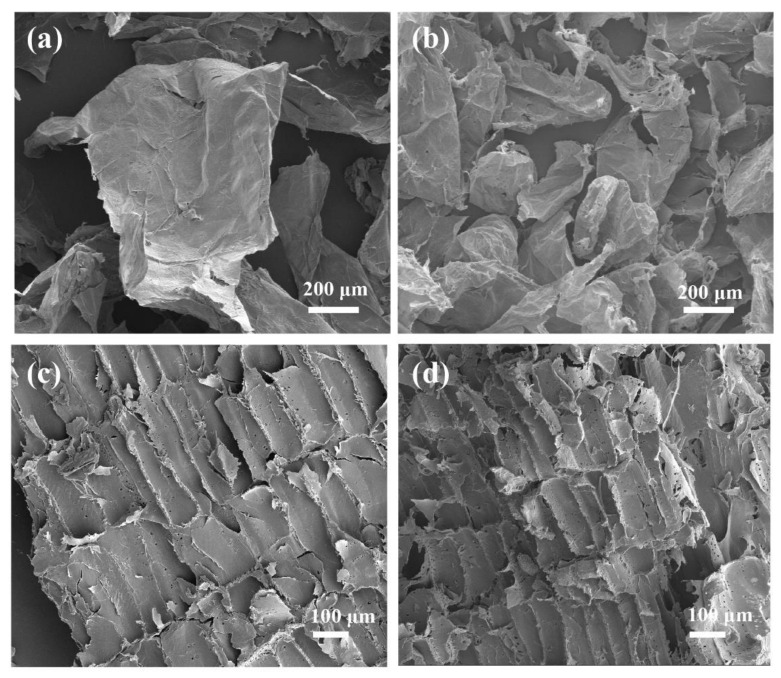
Morphology of different samples: (**a**) P-1, (**b**) P-4, (**c**) R-1 and (**d**) R-4.

**Figure 8 polymers-14-00930-f008:**
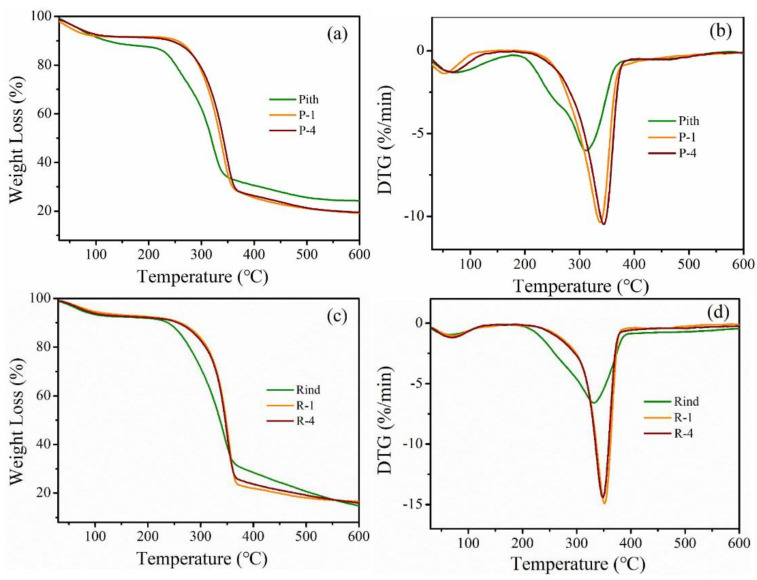
TGA curves of (**a**) pith and (**c**) rind with alkali treatments, DTG curves of (**b**) pith and (**d**) rind with alkali treatments.

**Figure 9 polymers-14-00930-f009:**
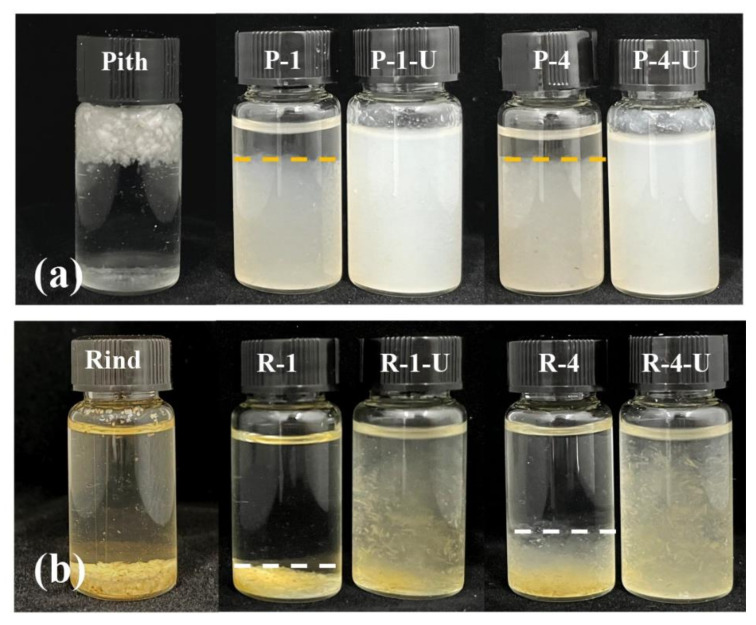
Photographs of suspensions of P-1-U and P-4-U (**a**), R-1-U and R-4-U (**b**) after 72 h of sedimentation.

**Figure 10 polymers-14-00930-f010:**
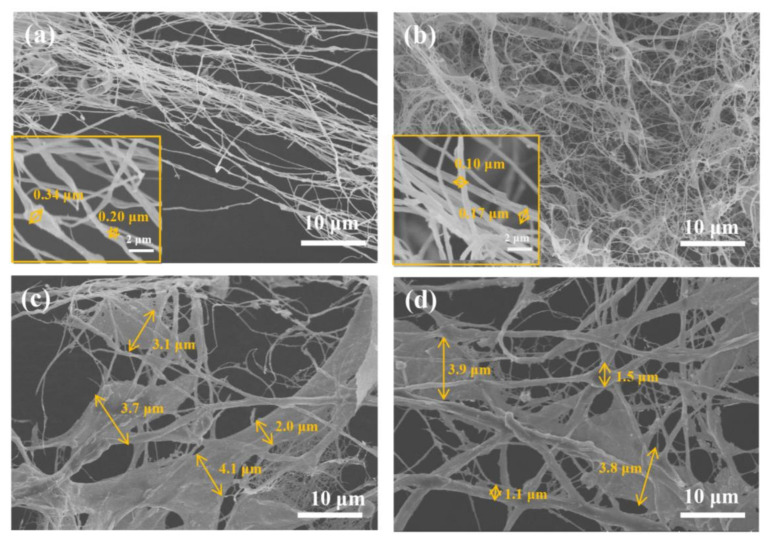
SEM mages of pith and rind after alkali treatment and high-intensity ultrasound. (**a**) P-1-U; (**b**) P-4-U; (**c**) R-1-U; (**d**) R-4-U.

**Figure 11 polymers-14-00930-f011:**
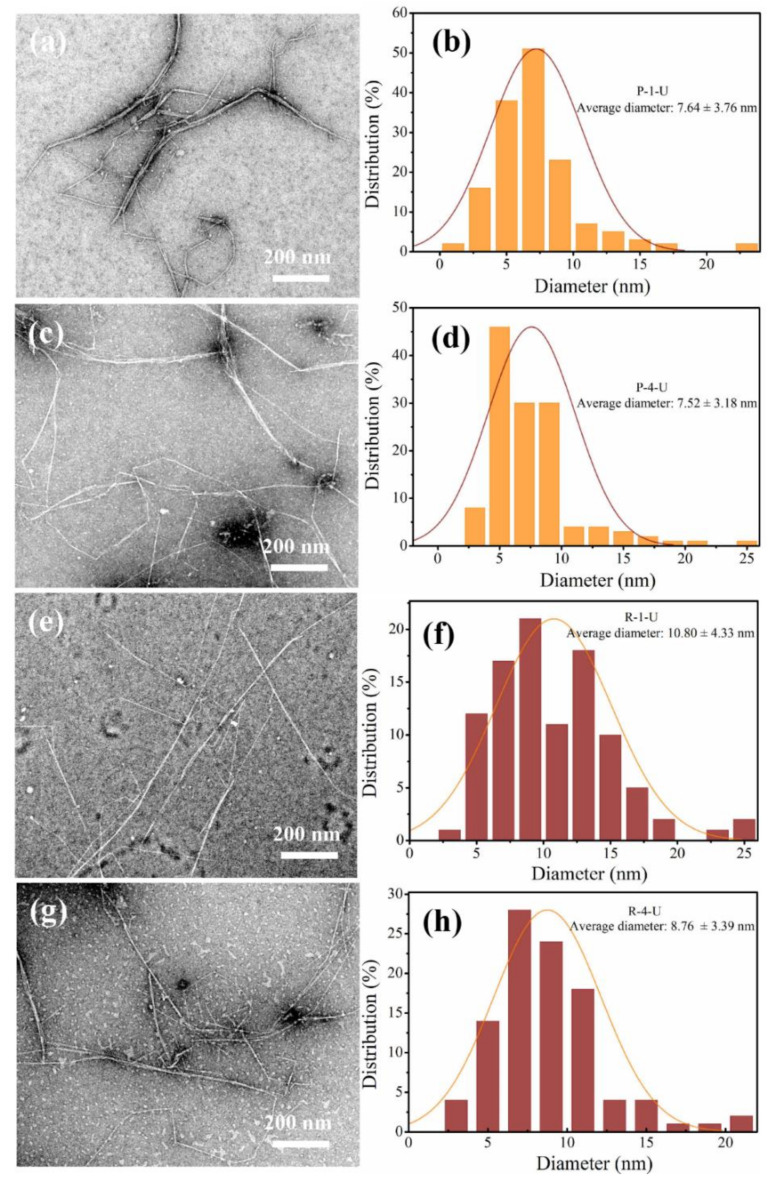
TEM images of the supernatant fraction of CNF suspension from pith (**a**,**c**) and rind (**e**,**g**) after 30-min of ultrasonication and their corresponding diameter distributions (**b**,**d**,**f**,**h**).

**Table 1 polymers-14-00930-t001:** Density, surface areas and pore volume of pith and rind.

Materials	Density (mg/cm^3^)	BET Surface Area (m^2^/g)	Pore Volume (cm^3^/g)
Pith	27 ± 3.22	1.77	0.006
Rind	318 ± 45.98	0.73	0.003

**Table 2 polymers-14-00930-t002:** Thermogravimetric properties of pith and rind with alkali treatments.

Materials	T_initial_ (°C)	T_30_ (°C)	T_max_ (°C)
Pith	221	281	311
P-1	255	313	338
P-4	261	318	344
Rind	246	304	331
R-1	270	329	351
R-4	275	331	349

T_initial_, temperature corresponding to initial weight loss; T_30_*,* temperature corresponding to 30% weight loss; T_max_, temperature corresponding to the maximum rate of weight loss.

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
