# Peer review of "A Comparative Investigation on Structural and Chemical Differences between the Pith and Rind of Sunflower Stalk and Their Influences on Nanofibrillation Efficiency"

_polymers, 2022, doi:10.3390/polym14050930_

Round 1

Reviewer 1 Report

  • The Materials section is incomplete. Bring all the materials information that was used in this research.
  • Section 2.2: full word of CNF is necessary first.
  • Page 4/14: explain “IUPAC classification”. What is “type II and type IV”?
  • Compare the graphs of FTIR with other similar studies and explain in the description.
  • What are P-1, P-4, R-1, and R-4? Explain how these four samples were added to the experiment?
  • Figure 8 is not explained clearly. (Explain about P-U and R-U).
  • Just as a suggestion the authors can add the future prospect at the end of the manuscript.
  • Reference 24 is too old, and it is better to be replaced with some newly published papers.
  • It is recommended to use the following references in the manuscript:

Sabbagh, F., Muhamad, I. I., PaLe, N., & Hashim, Z. (2018). Strategies in Improving Properties of Cellulose-Based Hydrogels for Smart Applications. Cellulose-Based Superabsorbent Hydrogels. Springer International Publishing, 887-908.

Sabbagh, F., & Muhamad, I. I. (2017). Physical and chemical characterisation of acrylamide-based hydrogels, Aam, Aam/NaCMC and Aam/NaCMC/MgO. Journal of Inorganic and Organometallic Polymers and Materials27(5), 1439-1449.

Reviewer 2 Report

This article is significance of content to some extent in the field of the nanofibrillation technology. However, I still suggest authors revise and improve this manuscript because there are some issues to address.

  • At the fourth line of the first paragraph in the second page, the authors described “The stalk consists of….”. Because you had explained where the pith and rind on the cross-sectional figure of sunflower stalk, please add the cross-sectional figure of sunflower stalk.
  • At the section 2.1 in the second page, the authors described “The sunflower stalk were separated….”. Please explain how to separate the pith and rind. Where is the line between the pith and rind?
  • In the fourth page, the figure 3a is missing in the article. Please add it.
  • In the Table 1, please explain how to measure the density and pore volume of the pith and rind.
  • In the figure 5, did you use the reference peak to normalize the absorbance in the all FTIR spectra? Generally, the absorbance is normalized to compare the differences before and after treatment. Otherwise, there is no way to account for an increase or decrease in absorbance. Additionally, before treatment, the peak for the pith and rind appeared at around 1250 cm-1. After treatment, they significantly decreased. Please add to explain this phenomenon.
  • The wrong typo “T30oC” is in the annotation of the table 2. Please revise it.
  • In conclusion, please add to describe how to improve nanofibrillation efficiency for the rind. What's your suggestion for a nanofibrillation method?
